# Prevalence and risk factors for multi-drug resistant *Escherichia coli* among poultry workers in the Federal Capital Territory, Abuja, Nigeria

**Mabel Kamweli Aworh**[1,2,3¤]*, **Jacob Kwaga**[3☉], **Emmanuel Okolocha**[3☉], **Nwando Mba**[4], **Siddhartha Thakur**[5☉]

1 Department of Veterinary and Pest Control Services, Federal Ministry of Agriculture and Rural Development, Abuja, Nigeria, 2 Nigeria Field Epidemiology and Laboratory Training Programme, Abuja, Nigeria, 3 Department of Veterinary Public Health and Preventive Medicine, Ahmadu Bello University, Zaria, Nigeria, 4 National Reference Laboratory, Nigeria Center for Disease Control, Abuja, Nigeria, 5 Department of Population Health and Pathobiology, College of Veterinary Medicine, North Carolina State University, Raleigh, North Carolina, United States of America

☉ These authors contributed equally to this work.
¤ Current address: Visiting Scholar, Department of Population Health & Pathobiology, College of Veterinary Medicine, North Carolina State University, Raleigh, North Carolina, United States of America
* mabelaworh@yahoo.com

**Data Availability Statement:** All relevant data are within the manuscript and its Supporting Information files.

## Abstract

### Background

Antimicrobial resistance has emerged as a global health threat. Antimicrobial resistant *Escherichia coli* infections are associated with high morbidity and expenditure when compared with infections caused by susceptible strains. In Nigeria, antimicrobial drugs are readily available over-the-counter with potential for indiscriminate use by poultry farmers and eventual development of drug resistance. The objective of this study was to investigate prevalence and risk factors for multi-drug resistant *E. coli* among poultry workers (PW) in Abuja, Nigeria.

### Materials and methods

A cross-sectional study was conducted among 122 randomly selected apparently healthy poultry workers (poultry-farmers/ sellers) in Municipal and Kuje Area Councils from December 2018 to April 2019. Data was collected on socio-demographics and exposure factors using a structured interviewer-administered questionnaire. *E. coli* was isolated and identified from stool samples of poultry workers. Antibiotic susceptibility testing was done using Kirby-Bauer disk diffusion method. Multidrug resistance (MDR) was defined as resistance to three or more classes of antimicrobials. Data was analyzed by computing proportions, prevalence odds-ratios (POR) and logistic regression at 5% significance level.

### Results

Among PW, there were 121 males (99.2%). Mean age of the male workers was 30.6 ± 9.7years, 54.6% (n = 66) married, 57.9% (n = 70) had secondary education and 62.0%

**Funding:** The authors received no specific funding for this work.

**Competing interests:** The authors have declared that no competing interests exist.

(n = 75) were farm-workers. Prevalence of *E. coli* was 39.7% (n = 48), highest among farm-workers (POR = 2.7, 95% Confidence Interval [CI] = 1.3–5.7; p = 0.01) compared to poultry-sellers. Of the 48 *E. coli* isolates, 16.7% (n = 8) were extended spectrum beta lactamase (ESBL) producers and 79.2% (n = 38) were MDR. We detected resistance against Tetracycline: (83.3%, n = 40), Sulfamethoxazole-Trimethoprim: (79.2%, n = 38), Ampicillin: (77.1%, n = 37), Streptomycin: (72.9%, n = 35), Nalidixic acid: (50%, n = 24), Gentamicin: (41.7%, n = 20), Chloramphenicol: (31.3%, n = 15), Cephalothin: (27.1%, n = 13), Nitrofurantoin: (10.4%, n = 5) and Imipenem: (6.3%, n = 3). Absence of lavatory (POR = 2.7, 95% CI = 1.1–6.7); existence of farm/market for >10years (POR = 2.5, 95% CI = 1.1–5.4) and PW's history of diarrhea in last three months (POR = 2.8, CI = 1.2–6.3) were associated with MDR. Controlling for age, absence of lavatory (adjusted OR [aOR] = 4.31, 95% CI = 1.6–11.9); PW's history of diarrhea in last three months (aOR = 3.3,95%CI = 1.3–8.5) and work exposure >10years (aOR = 0.3, 95%CI = 0.1–0.9) remained independent risk factors for MDR.

## Conclusion

Prevalence of resistant *E. coli* was highest among farm-workers and associated with older farms/markets, occupational exposure of over 10 years and poor hygienic measures. The management of Municipal and Kuje Area Councils were recommended to provide lavatories for public use in farm-settlements/markets. The importance of hand-hygiene and responsible use of antimicrobials in poultry production was emphasized.

## Introduction

Globally, *Escherichia coli* is an important cause of food borne illness and a public health threat, especially in sub-Saharan Africa including Nigeria [1]. Antimicrobial resistance (AMR) has recently gained worldwide recognition as the emergence of multidrug resistant organisms has led to increased mortality and economic burden, and Nigeria is no exception to the challenges faced due to AMR [1,2]. The increasing occurrence of drug-resistant *E. coli* isolates of human and animal origin is a global public health problem. Antibiotic resistant *E. coli* infections are presently associated with higher morbidity, mortality, and invariably higher expenditure in treatment compared with infections by strains susceptible to the drugs. Drug resistant commensal *E. coli*, which may not directly cause disease, remains significant in public health as a reservoir of drug resistance genes. These genes can be transferred to humans and zoonotic pathogens, such as *E. coli*, *Salmonella*, or to other gram-negative bacteria in the gut [3,4].

In developing countries including Nigeria, many families depend on poultry production as a means of income and livelihood due to the increased consumption of poultry products [5]. Antimicrobial drugs are also readily available over the counter without prescription because access to veterinary drugs is presently not being regulated in the country, thereby encouraging the use of these drugs by poultry farmers indiscriminately in production. Poultry farmers continue to use antibiotics in poultry feed or water for prophylaxis, treatment of diseases and as growth promoters in Nigeria [6,7]. Commonly used antibiotics in poultry production in Nigeria include; oxytetracycline, neomycin, enrofloxacin, doxycycline, gentamicin, colistin, streptomycin, tylosin, ciprofloxacin, nitrofurans and chloramphenicol. This is similar to the practice in most countries [7].

In 2018, Nigeria's National Agency for Food and Drugs Administration and Control (NAFDAC) issued a ban on the use of antibiotics as growth promoters in animal feeds.

Studies have shown that one of the risk factors for the acquisition of extended spectrum beta lactamase (ESBL)—producing organisms is the abuse of antibiotics in poultry production [8]. Resistance to commonly used antibiotics has major socioeconomic and public health implications. The socioeconomic implications of AMR include increased cost and duration of treatment while the public health implications include decreased ability to treat common infections resulting in increased human suffering and ultimately death [9–11].

This study focused on the prevalence of multi-drug resistant *E. coli* in humans working on poultry farms and live bird markets (LBM). Globally, there have been an increase in the challenges related to treatment of human and animal bacterial infections attributed to the development of antimicrobial resistance [12]. The possibility of antimicrobial resistance genes circulating among humans, animals and the environment constitutes a direct threat to public health. This is why it is critical to develop new approaches and institute strict control measures in accessing and using antimicrobials in humans and animals [13]. Antimicrobial use and resistance situation analysis conducted by Nigeria in 2017 in response to the 68th World Health Assembly (WHA) resolution 68.7 identified some data gaps with regards to studies linking AMR in humans, food animals and the environment. Systematic review of previous studies showed that studies evaluating resistance patterns of *E. coli* in Nigeria had a skewed distribution, with a higher concentration of such studies carried out in the South West and none in the North East and South Eastern parts of the country [14].

Although, few studies have evaluated MDR *E. coli* in Nigeria, most available data are specific to strains that are pathogenic either to humans or animals [15,16]. The increasing use of antimicrobials in livestock and poultry production in developing countries such as Nigeria may increasingly lead to development of AMR among commensals and enteric pathogens such as *E. coli*. This occurrence and monitoring of AMR among *E. coli*, which has become a model organism for surveillance has become imperative. Poultry is a potential source of MDR *E. coli* with the potentials to spread of AMR genes to other bacteria in the environment and human populations in Nigeria. Data on the available antimicrobials that these bacterial organisms are already resistant to is required for proper diagnosis and treatment of bacterial infections in humans.

We hypothesized that poultry harboring drug resistant *E. coli* aids in the transmission of the pathogen to human workers who are exposed in farms and live bird markets. The objective of this study was to determine the prevalence and risk factors for MDR *E. coli* among poultry workers in the Federal Capital Territory (FCT), Abuja Nigeria to generate baseline data for the implementation of the National Action Plan on AMR in Nigeria.

## Materials and methods

### Study area

The Federal Capital Territory (FCT), Abuja, North Central Nigeria is a cosmopolitan city with a land area of 8,000 square kilometers. It has six Area Councils (ACs) for administrative purposes–Abuja Municipal Area Council (AMAC), Kuje, Abaji, Kwali, Gwagwalada and Bwari. Based on data obtained from the National Bureau of Statistics, the population projection for FCT for 2019 is 4,464,785 people [17]. This study was conducted in Abuja Municipal Area Council (9.0612˚ N, 7.4224˚ E) and Kuje Area Council (8˚ 52' 46.27" N, 7˚ 13' 39.22" E) with the largest poultry population of approximately 3,812,288 birds [18].

## Study design

We conducted a cross-sectional study among apparently healthy poultry workers who were randomly selected from farms and LBMs situated in Abuja Municipal and Kuje Area Councils from December 2018 to April 2019. All apparently healthy poultry workers who were 18yrs and above present at the farm/LBM at time of study were included in the study. We collected data on socio-demographics and exposure factors using a structured interviewer-administered questionnaire. The electronic questionnaire, designed using open data kit (ODK) software and installed on a smart phone, was administered by the researcher on all the study participants. Individual stool samples were collected from all the workers in sterilized stool collection containers. All respondents were given an informed consent form detailing the purpose of the study which was signed by all who agreed to participate in the study prior to sample collection. Ethical approval for this study was obtained from the Scientific and Ethical committee of the FCT Health Research Ethics Committee in July 2018 (Approval Number: FHREC/2018/01/84/16-07-18). Permission was sought from the management of each study site prior to commencement of study. Confidentiality of information obtained was assured.

## Sampling method

AMAC and Kuje Area Councils were selected based on the level of commercial poultry production comprising of layers, broilers and local indigenous chickens. The layers are exotic improved breeds and provide majority of the table eggs available for consumption [18]. It has been documented that over 3,812,288 chickens are present in FCT, Abuja out of which 16% are exotic improved breeds [18]. A systematic sampling method was used to select farms/LBMs based on a list of registered farms/LBMs provided by the FCT Department of Agriculture and Rural Development. A total of 52 poultry farms were randomly selected from the list of farms provided for the two Area Councils (22 farms from AMAC and 30 farms from Kuje). AMAC has more than six (6) LBMs while Kuje AC has only one LBM. Seven LBMs were selected in AMAC and one in Kuje for the study. A total of 122 apparently healthy poultry workers were enrolled for the study. Freshly passed stool samples (one sample per poultry farmer/ poultry seller) were collected from randomly selected poultry workers in sterile stool containers and transported in cool boxes to the National Reference Laboratory, Nigeria Centre for Disease Control Gaduwa, Abuja. The stool samples were processed within 3hours of sample collection for the presence of *E. coli*. A farm or LBM was considered to be infected and/or contaminated when at least one stool sample tested positive for *E. coli*.

## Questionnaire survey

The determination of risk factors for *E. coli* infections involved the administration of questionnaires to poultry farmers/ LBM sellers to assess the association between management practices, general hygiene and acquiring *E. coli* infections (S1 Appendix). Permission was sought from farm/LBM management prior to administering the questionnaire. Demographic and management data collected include farm house characteristics, management of ill birds, rodent control, presence of other domestic animals on farm, feeding and watering practices, farm staff and visitors, cleaning and disinfection procedures. The questionnaire was pretested on 5 farms in and around Zaria however, data from the pretesting was not included in the final analysis.

## Bacteriological analysis

**Isolation and identification of *E. coli* isolates. Enrichment of the samples**. Briefly about one gram of human stool sample was inoculated in enrichment broth (buffered peptone

water) and incubated at 37˚C for 24 h. Thereafter, a loop-full culture from enrichment broth was streaked onto MacConkey's lactose agar and incubated at 37˚C for 24 h. Suspected *E. coli* colonies, usually pink to red were picked and further streaked on Eosin Methylene Blue (EMB) agar [19].

**Plating**. Three well-isolated suspected *E. coli* colonies usually pink in colour were selected at random from the MacConkey lactose agar plate and then sub-cultured on to Eosin Methylene Blue (EMB) agar, with a sterile inoculating loop and incubated at 37˚C under aerobic conditions for 24h. Colonies that are raised, moist with greenish metallic sheen, suggestive of *E. coli*, were then sub-cultured onto trypticase soy agar plates and incubated for 24h at 37˚C under aerobic conditions for the isolation of pure cultures. All *E. coli* isolates were further screened using conventional biochemical tests (Triple sugar iron, Urease, Sulphur Indole motility, Methyl red, Voges–Proskauer, and Citrate utilization biochemical tests) [19]. Isolates tentatively shown to be *E. coli* in the conventional biochemical screening tests were subjected to further tests using commercially available biochemical test strip, Microbact GNB 24E (Oxoid, UK), for confirmation. This kit was used according to the Manufacturer's instructions.

**Antimicrobial susceptibility profiling.** The antibiotic susceptibility patterns of *E. coli* isolates were tested using the Kirby Bauer disk diffusion method [20] and according to the Clinical and Laboratory Standards Institute guidelines (CLSI, 2018). Briefly, one colony of the test isolates from overnight cultures grown on trypticase soy agar was picked using sterile Pasteur loop and emulsified in sterile normal saline. The turbidity of the suspension was adjusted to 0.5 MacFarland's standard and then using sterile cotton swab, the bacteria was spread on Mueller Hinton agar to obtain a lawn culture. After air drying, commercially available antibiotic discs (Oxoid, UK) were placed 30mm apart and 10mm away from the edge of the plate and incubated at 37˚C for 24hours. Inhibition zone diameter was measured, recorded and values interpreted using standard recommendations of the Clinical and Laboratory Standards Institute (CLSI, 2018). The isolates were tested using a panel of 16 antibiotics of different classes commonly used to treat human and animal bacterial infections namely ampicillin (10μg), amoxycillin/ clavulanic acid (20/10μg), tetracycline (30μg), gentamicin (10μg), cefuroxime (30μg), streptomycin (10μg), chloramphenicol (30μg), nalidixic acid (30 μg), sulfamethoxazole-trimethoprim (10μg), cephalothin (30μg), nitrofurantoin (300μg), ceftriaxone (30μg), imipenem (10μg), colistin (10μg), ceftazidime (30μg) and cefotaxime (30μg). A standard reference strain of *E. coli* (ATCC 25922), sensitive to all antimicrobial drugs being tested, was used as a control. We defined multi-drug resistance (MDR) as resistance to three or more classes of antimicrobials.

**Determination of multiple antibiotic resistance (MAR) index.** The multiple antibiotic resistance (MAR) index for each isolate was determined using the formula MAR = a/b, where *'a'* is the number of antibiotics to which the test isolate was resistant to and *'b'* is the total number of antibiotics to which the test isolates were subjected [21].

**Detection of ESBL phenotype by double disk methods.** All *E. coli* isolates were screened for the production of ESBLs by using double disk method as described by [22]. From the pure cultures of bacteria grown overnight on MacConkey agar, a suspension matching 0.5 McFarland standard (1.5 x 108 CFU/ml) was made in normal saline. Using sterile cotton swab, the bacteria was spread on Mueller Hinton agar to obtain a lawn culture. After allowing the plate to dry, the antibiotic disk amoxycillin/clavulanic acid 20+10 mg was placed at the center of the plate. Next, one cefotaxime 30 mg disk and disk of ceftazidime 30 mg were placed 15mm apart from the edge of amoxycillin/ clavulanic acid disk. The plates were incubated in air at 37˚C for 16–18 hours. Following growth, the diameter of the zones of inhibition around the disks were measured and recorded. Any *E. coli* isolate that showed an increase in the zone of inhibition

around either ceftazidime or cefotaxime (i.e., > 5 mm towards the disc of amoxicillin-clavulanate) was interpreted as positive for the ESBL production. *E. coli* ATCC 25929 and *K. pneumoniae* ATCC 700603 were used as negative and positive controls, respectively.

## Statistical analysis

Data collected were entered into ODK collect on a smart phone during the interview and skip logics was used to limit wrong data entries. Data were extracted from the smart phone using ODK briefcase, converted to MS Excel format then cleaned and analyzed using Epi Info version 7 software. Data was analyzed by computing proportions and prevalence odds-ratios (POR). Factors found to be significant at the bivariate analysis were included in the logistic regression model for multivariate analysis at 5% level of significance.

## Results

Of the 122 poultry workers we sampled, 99.2% (n = 121) were male. The analysis presented in the results section was based on 121 males whose ages ranged from 18 to 67 years. The mean age of study participants was 30.6 ± 9.7years. The majority of workers were married: 54.6% (n = 66), had secondary education: 57.9% (n = 70) and worked on a poultry farm: 62.0% (n = 75) (Table 1).

### Prevalence of multidrug resistant *E. coli* among poultry workers

A total of 48 (39.7%) individuals tested positive for MDR *E. coli*. A majority of these, 62.5% (n = 30) resided in Abuja Municipal Area Council (Fig 1). The highest prevalence was among farm-workers (POR = 2.7, 95% Confidence Interval [CI] = 1.3–5.7; p = 0.01) when compared to poultry-sellers.

Of 48 isolates, 16.7% (n = 8) were ESBL producers; 79.2% (n = 38) were MDR; resistance was detected against tetracycline (83.3%, n = 40), sulfamethoxazole-trimethoprim: (79.2%,

**Table 1. Socio-demographic characteristics of poultry workers on Farms and Live Bird Markets in FCT, Abuja—Nigeria, 2019.**

| Characteristics | n | % |
|---|---|---|
| **Marital Status**: | | |
| Married | 66 | 54.6 |
| Single | 55 | 45.5 |
| **Educational level**: | | |
| Islamic education | 10 | 8.3 |
| Primary | 19 | 15.7 |
| Secondary | 70 | 57.9 |
| Tertiary | 22 | 18.2 |
| **Profession**: | | |
| Poultry farm worker | 75 | 62.0 |
| Poultry seller | 46 | 38. |
| **Duration of work**: | | |
| 0–9 years | 83 | 68.6 |
| ≥10 years | 38 | 31.4 |
| **Location**: | | |
| AMAC | 69 | 57.0 |
| Kuje | 52 | 43.0 |

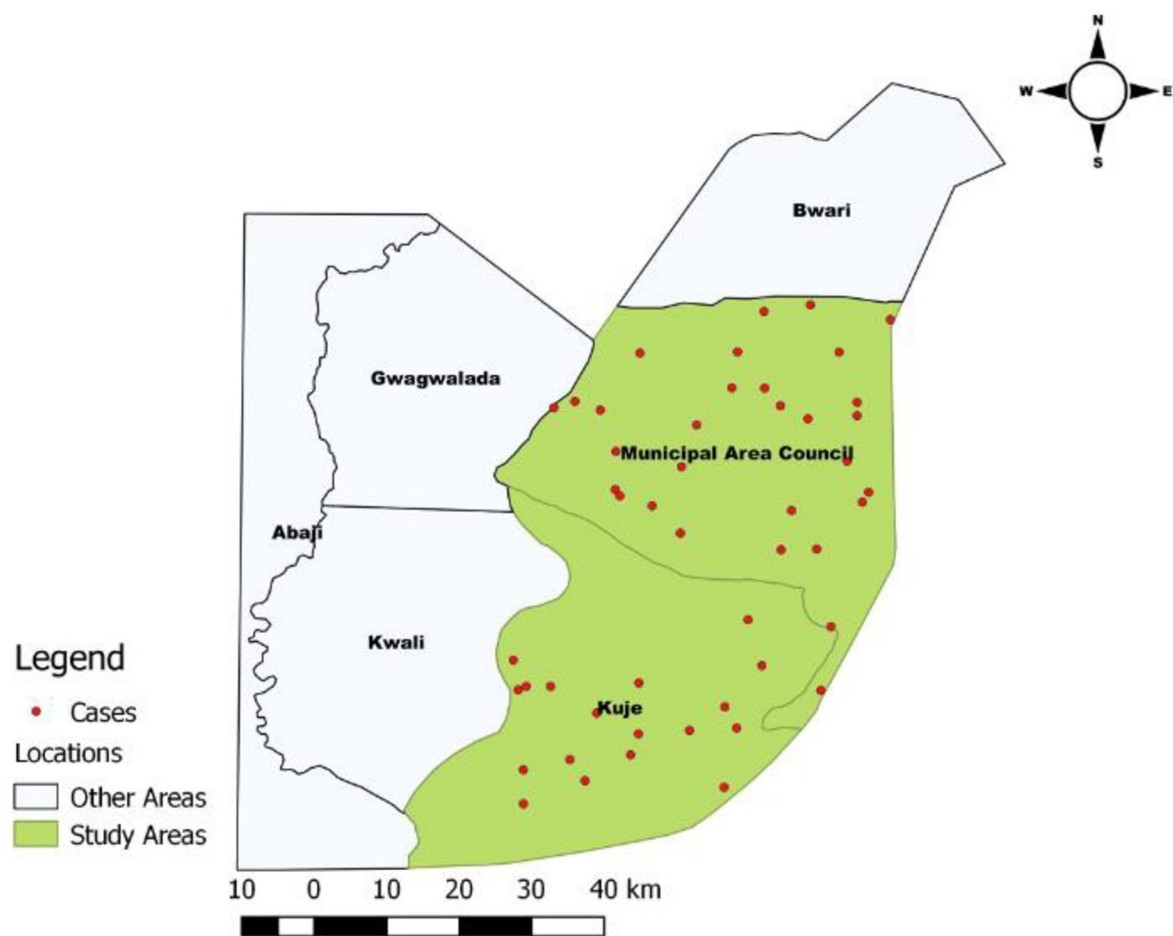

**Fig 1. Spatial distribution of *E. coli* positive isolates among poultry workers in FCT, Abuja, Nigeria– 2019.**

n = 38), ampicillin (77.1%, n = 37), streptomycin (72.9%, n = 35), nalidixic acid: (50%, n = 24), gentamicin: (41.7%, n = 20), chloramphenicol: (31.3%, n = 15), cephalothin: (27.1%, n = 13), nitrofurantoin: (10.4%, n = 5) and imipenem: (6.3%, n = 3) see Fig 2. Among ESBL producers, the maximum number was isolated from stool samples of poultry-sellers: 75% (n = 6) when compared to farm-workers.

## Antimicrobial resistance patterns of *E. coli* Isolates

Overall, colistin (100%), amoxicillin-clavulanate (95.8%), imipenem (93.8%), ceftriaxone (93.8%), ceftazidime (91.7%), cefotaxime (91.7%), cefuroxime (91.7%) and nitrofuratoin (89.6%) were found to be susceptible to *E. coli* (Table 2).

The resistance rates of isolates to tetracycline, sulfamethoxazole-trimethoprim, ampicillin, streptomycin, nalidixic acid, gentamicin, chloramphenicol, cephalothin and nitrofurantoin were much higher among farm-workers when compared to poultry-sellers. However, resistance rates of isolates to the third and fourth generation cephalosporins (ceftriaxone, cefuroxime, cefotaxime and ceftazidime) were quite low and observed only among poultry-sellers at live bird markets (Table 2).

This study observed that 81.3% (n = 39) of the isolates had multiple antibiotic resistance (MAR) index greater than 0.2 while 18.8% (n = 9) isolates had MAR index less than 0.2 (Fig 3).

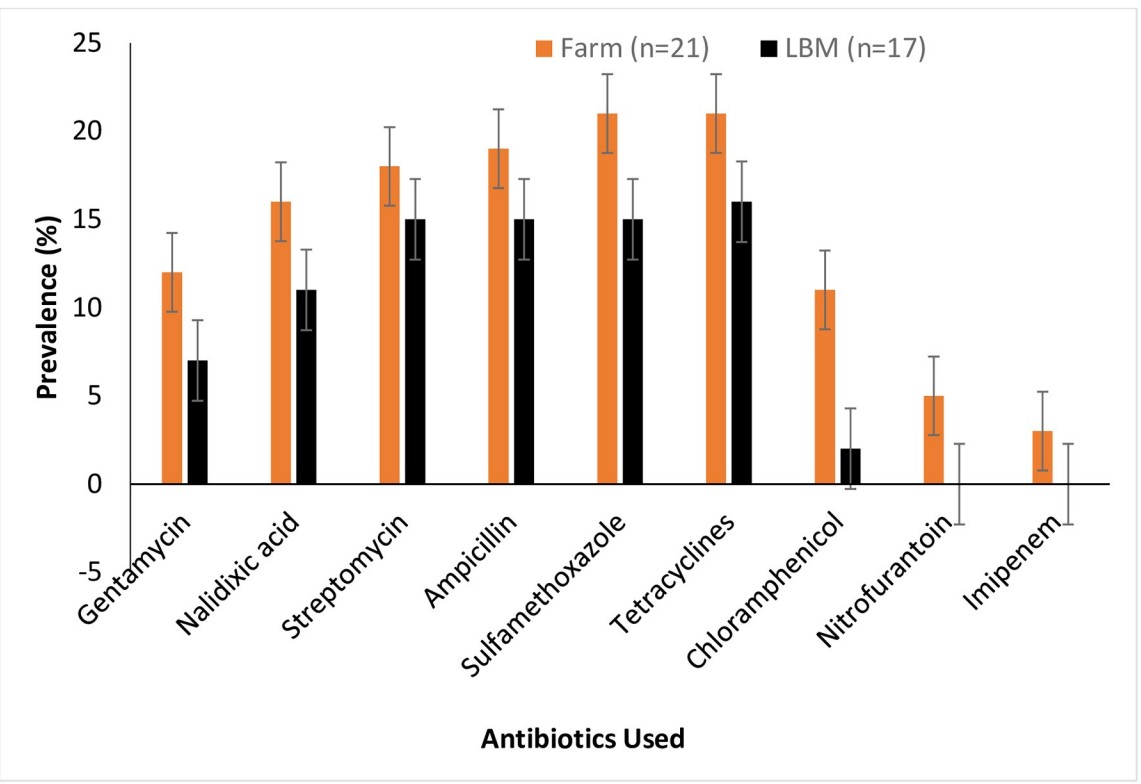

**Fig 2. Prevalence of MDR *E. coli* isolated from stool samples of poultry workers on farms and live bird markets in FCT, Abuja, Nigeria– 2019.** Bars represent the proportion of samples containing at least one antibiotic-resistant E. coli isolate with 95% confidence intervals. Error bars represent Standard Error of the mean prevalence. Data was obtained from two sources: poultry farms and live bird markets.

Of the eight ESBL producing *E. coli* isolated from poultry workers, we recorded high resistance for tetracycline: 100% (8/8) followed by streptomycin: 88% (7/8), sulfamethoxazole-trimethoprim: 75% (6/8), ampicillin: 75% (6/8), nalidixic acid: 63% (5/8), gentamicin: 38% (3/8), chloramphenicol: 38% (3/8) and cephalothin: 38% (3/8).

### Risk factors for multidrug resistant *E. coli* among poultry workers

Factors associated with MDR *E. coli* among poultry workers in the FCT, Abuja were: absence of lavatory (POR = 2.7, 95% CI = 1.1–6.7); existence of farm/market for over 10years (POR = 2.5, 95% CI = 1.1–5.4); poultry worker's history of diarrhea in last three months (POR = 2.8, CI = 1.2–6.3) and work exposure over 10years (POR = 2.1, 95% CI = 1.0–4.8), see Table 3.

In the logistic regression model used, the following factors were significant at bivariate analysis at p ≤ 0.05 and subsequently added in the model, including absence of lavatory, existence of farm/market for over 10years, work exposure of over 10years and poultry worker's history of diarrhea in the last three months. After controlling for age and using a stepwise elimination approach, only three factors remained statistically significant in the logistic regression model. In the final logistic regression model, absence of lavatory on the poultry farm (adjusted OR [aOR] = 4.31, 95% CI = 1.6–11.9); poultry worker's history of diarrhea in last three months (aOR = 3.3, 95%CI = 1.3–8.5) and work exposure of over 10years (aOR = 0.3, 95%CI = 0.1–0.9) remained independent risk factors for getting infected with MDR *E. coli* among poultry

**Table 2. Frequency of resistance to the antimicrobials tested by *Escherichia coli* isolates from poultry workers, on Farms and Live Bird Markets in FCT, Abuja, Nigeria– 2019.**

| Drug Class | Drug | Resistance break point μg/mL | Overall, n = 122 (%) | Poultry Farms n = 22 (%) | Live Bird Markets n = 26 (%) |
|---|---|---|---|---|---|
| **Tetracyclines** | Tetracycline (30μg) | ≤ 11 | 40 (83.3%) | 21 (95.5%) | 19 (73.1%) |
| **Folate Pathway antagonists** | Sulfamethoxazole Trimethoprim (10μg) | ≤ 10 | 38 (79.2%) | 21 (95.5%) | 17 (65.4%) |
| **Penicillins** | Ampicillin (10μg) | ≤ 13 | 37 (77.1%) | 20 (90.9%) | 17 (65.4%) |
| **Quinolones** | Nalidixic acid (30μg) | ≤ 13 | 24 (50.0%) | 15 (68.2%) | 9 (34.6%) |
| **Aminoglycosides** | Streptomycin (10μg) | ≤ 11 | 35 (72.9%) | 18 (81.8%) | 17 (65.4%) |
| | Gentamicin (10μg) | ≤ 12 | 20 (41.7%) | 12 (54.6%) | 8 (30.8%) |
| **Phenicols** | Chloramphenicol (30μg) | ≤ 12 | 15 (31.3%) | 11 (50.0%) | 4 (15.4%) |
| **1st Generation Cephalosporins** | Cephalothin (30μg) | ≤ 14 | 13 (27.1%) | 5 (22.7%) | 8 (30.8%) |
| **Nitrofurans** | Nitrofurantoin (300μg) | ≤ 14 | 5 (10.4%) | 4 (18.2%) | 1 (3.9%) |
| **Carbapenems** | Imipenem (10μg) | ≤ 19 | 3 (6.3%) | 3 (13.6%) | 0 (0%) |
| **B-lactam inhibitors** | Amoxicillin-clavulanate (20/10μg) | ≤ 13 | 2 (4.2%) | 2 (9.1%) | 0 (0%) |
| **3rd and 4th Generation Cephalosporins** | Ceftriaxone (30μg) | ≤ 19 | 3 (6.3%) | 0 (0%) | 3 (11.5%) |
| | Cefuroxime (30μg) | ≤ 14 | 4 (8.3%) | 0 (0%) | 4 (15.4%) |
| | Cefotaxime (30μg) | ≤ 22 | 4 (8.3%) | 0 (0%) | 4 (15.4%) |
| | Ceftazidime (30μg) | ≤ 17 | 4 (8.3%) | 0 (0%) | 4 (15.4%) |
| **Resistance to 3 or more classes of antimicrobials** | MDR | n/a | 38 (7.2%) | 21 (95.5%) | 17 (65.4%) |

workers in FCT, Abuja (Table 4). Based on logistic regression analysis, we found that poultry workers were significantly more likely to shed antimicrobial-resistant *E. coli* if they did not have access to toilets (p = 0.01) or had reported to have diarrhea in past 3 months (p = 0.02) or had been exposed to poultry either on farms / LBMs for over 10 years (p = 0.04).

## Discussion

To our knowledge, information on MDR *E. coli* among poultry-farmers and poultry-sellers in Nigeria is not available. Therefore, in this study, we examined the prevalence and risk factors for MDR *E. coli* isolates from apparently healthy poultry-farmers and poultry-sellers. This study showed that prevalence of MDR *E. coli* among poultry workers was high especially among farm-workers in the two area councils with the highest production and sale of poultry in FCT, Abuja, Nigeria. This could be attributed to the fact that there is an established link between the use of antimicrobials in animals and the development of bacterial resistance in humans to these drugs [4,23]. A study done in Korea among poultry farm workers reported that 43% of *E. coli* isolated from the workers showed resistance to four or more antimicrobials used in poultry production [24]. Similarly another study in the Netherlands reported that 27% of *E. coli* isolated from broiler and layer chicken farmers showed resistance to more than three antimicrobials [25].

It has been documented that multiple antibiotic resistance is mostly observed by the action of multidrug efflux pumps, each of which can pump out more than one drug type [2]. Our findings showed that 81.3% of *E. coli* isolated from poultry workers had a MAR index above 0.2. Studies have shown that MAR indices greater than 0.2 implies isolates from high-risk contaminated sources with frequency of antibiotic use [26]. This may be correlated with the indiscriminate use of antimicrobials in poultry production as prophylaxis and growth promotion purposes.

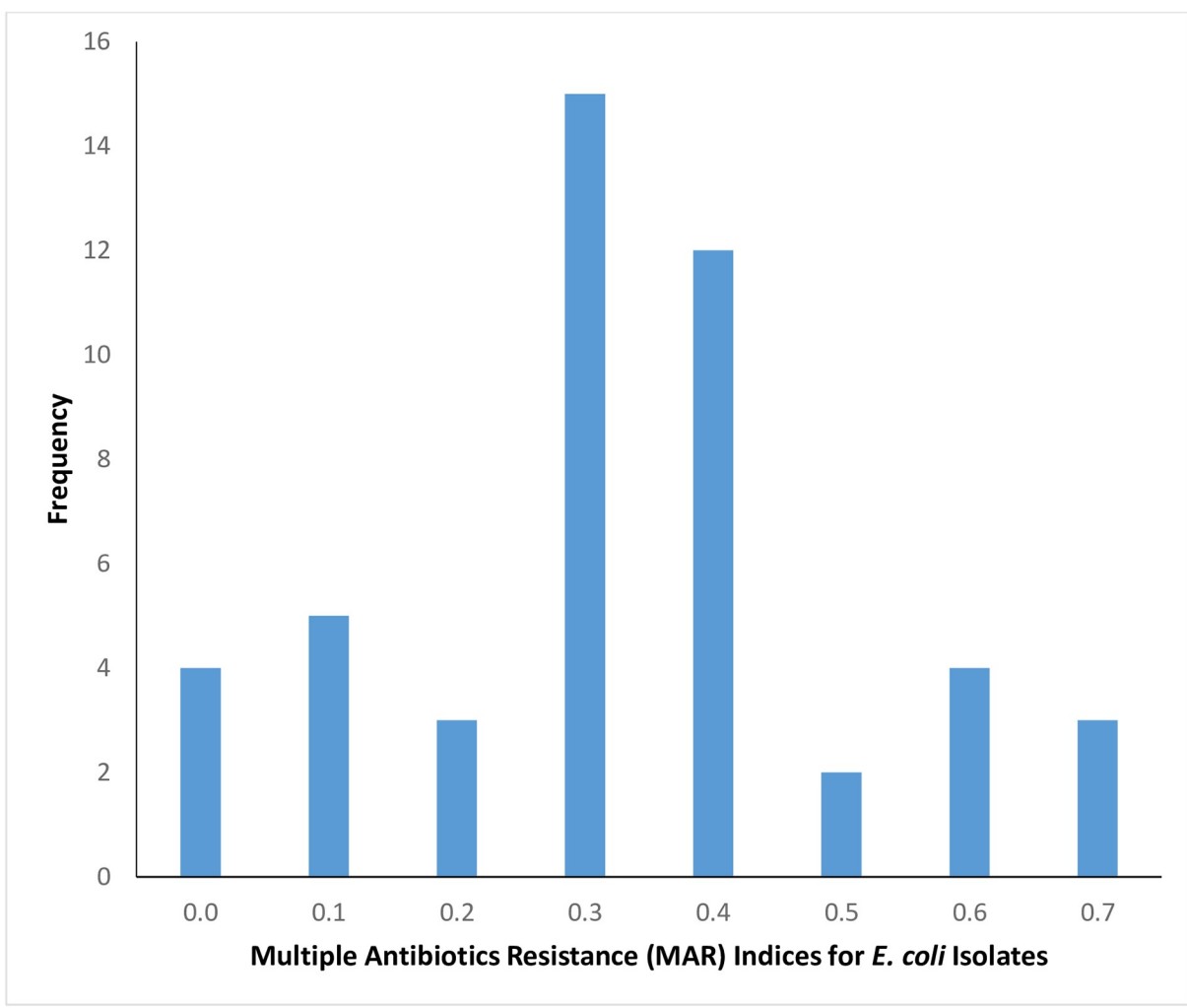

**Fig 3. Multiple antibiotics resistance indices for *E. coli* isolated from poultry workers on Farms and Live Bird Markets in FCT, Abuja, Nigeria– 2019.**

Our present study observed high prevalence of multidrug resistant *E. coli* but a low prevalence of ESBL *E. coli* in the study population. This is similar to the findings from a survey of households and chicken farms in the Mekong Delta in Vietnam which reported a high prevalence of multidrug resistant *E. coli* (81.3%), however a low prevalence of ESBL producing *E. coli* (3.2%) [27]. This may be correlated to the poor hygienic measures observed in our study hence the need to enlighten farmers and chicken sellers on the importance of hand hygiene on poultry farms/live bird markets.

Studies have shown that extended-spectrum β-lactamases (ESBL)—producing *E. coli* are resistant to several antibiotics especially penicillins and cephalosporins however, they are susceptible to cephamycins and carbapenems [28]. In our study, resistance rates to the third and fourth generation cephalosporins were quite low and observed only among poultry-sellers at live bird markets. It has been documented that cephalosporin resistant *E. coli* have been transmitted from food animals to humans and this supports our findings [29]. This is possibly because being a poultry-seller increases your risk to get infected compared to being a poultry farm worker as evident in our study.

**Table 3. Factors associated with multidrug resistant *E. coli* isolated from poultry workers on Farms and Live Bird Markets in FCT, Abuja, Nigeria– 2019 at bivariate analysis.**

| Work Exposure | MDR (Yes) | MDR (No) | Prevalence Odds Ratio (95% Confidence level) | P-value |
|---|---|---|---|---|
| **Age** | | | | |
| 18–29 years | 20 | 48 | 1.2 (0.6–2.6) | 0.64 |
| ≥ 30 years | 18 | 36 | | |
| **Level of Education** | | | | |
| Less than Secondary | 10 | 19 | 0.8 (0.3–2.0) | 0.66 |
| Secondary and above | 28 | 65 | | |
| **Profession** | | | | |
| Poultry farm worker | 22 | 55 | 1.4 (0.6–3.1) | 0.42 |
| Poultry seller | 16 | 29 | | |
| **Work Exposure** | | | | |
| ≥ 10 years | 17 | 23 | 2.1 (1.0–4.8) | **0.05*** |
| < 10 years | 21 | 61 | | |
| **Age of farm** | | | | |
| ≥ 10 years | 22 | 30 | 2.5 (1.1–4.9) | **0.02*** |
| < 10 years | 16 | 54 | | |
| **Source of drinking water** | | | | |
| Bad source | 15 | 22 | 1.8 (0.8–4.1) | 0.14 |
| Good source | 23 | 62 | | |
| **Absence of Lavatory** | | | | |
| Yes | 24 | 69 | 2.7 (1.1–6.4) | **0.02*** |
| No | | | | |
| **Rodent control** | | | | |
| No | 19 | 35 | 1.4 (0.7–3.0) | 0.39 |
| Yes | 19 | 49 | | |
| **Diarrhea in last 3 months** | | | | |
| Yes | 14 | 15 | 0.4 (0.2–0.9) | **0.02*** |
| No | 24 | 69 | | |
| **Waste Management** | | | | |
| On farm disposal | 21 | 54 | 1.5 (0.7–3.2) | 0.34 |
| Off farm disposal | 17 | 30 | | |

*Values that were significant at bivariate analysis.

**Table 4. Factors associated with multidrug resistant *E. coli* isolated from poultry workers on Farms and Live Bird Markets in FCT, Abuja, Nigeria– 2019 in logistic regression model.**

| Work Exposure Factors | Adjusted Odds Ratio | 95% Confidence interval | P value |
|---|---|---|---|
| Absence of lavatory | 4.31 | 1.57–11.89 | **0.01*** |
| Reported diarrhea in last 3 months | 3.25 | 1.25–8.50 | **0.02*** |
| Age of Farm/LBM | 0.48 | 0.19–1.19 | 0.11 |
| Work exposure of over 10 years | 0.26 | 0.07–0.92 | **0.04*** |

*Values that remained significant in the logistic regression model. In the final logistic regression model used for the multivariate analysis, the age group of the poultry workers was included

Findings from this study showed high frequency of tetracycline resistance and this is consistent with findings of other studies which explained that tetracycline is one of the most frequently used antibiotics in poultry production because they display a wide spectrum of antimicrobial action [30,31]. Our study also observed that ESBL producing *E. coli* isolates from poultry workers showed high resistance rates for tetracycline, streptomycin, sulfamethoxazole-trimethoprim, ampicillin and nalidixic acid, however, we observed low resistance rates for gentamicin, chloramphenicol and cephalothin. A similar study done in China among pig farmers reported high resistance rates for tetracycline, nalidixic acid and low rates for chloramphenicol [31]. A possible explanation for MDR *E. coli* observed among the poultry workers could be as a result of spread of antimicrobial resistance genes from poultry to the workers especially those with occupational exposure of over 10 years.

It has been documented that the mechanism of spread of antibiotic resistance from animals to humans is not clear cut however studies have reported that animals are a reservoir for *E. coli* found in humans [32]. Furthermore, it has been reported that antimicrobial resistance plasmids in *E. coli isolated* from chickens can spread to human handlers [33]. Studies have reported that food producing animals of which poultry is a part of are important carriers of ESBL producing *E. coli* [34]. This study, however, recorded a low rate of ESBL-producing *E. coli* among poultry workers. Other studies have reported that there is a connection between ESBL-producing *E. coli* in apparently healthy humans and those isolated from poultry [35].

The findings of this study showed a high prevalence of MDR *E. coli* isolates among poultry workers with isolates being resistant to tetracycline, sulfamethoxazole-trimethoprim, ampicillin and streptomycin. This is consistent with findings from a study done in Tanzania where *E. coli* isolates were reported to be resistant to ampicillin, tetracycline, sulfamethoxazole-trimethoprim or streptomycin [36].

Multidrug resistance to commonly used antibiotics observed in this study may be attributed to the indiscriminate use of antibiotics in poultry production in Nigeria especially since these drugs are easily accessible to poultry-farmers. Drug resistant commensal *E. coli* remains a reservoir of drug resistance genes and may also be responsible for the resistance pattern observed in our study [3].

Studies have reported that indiscriminate use of antimicrobial drugs in humans and animals has been responsible for the selective increase in certain bacterial populations [13,37]. Antimicrobial use and misuse promotes the emergence, selection and dissemination of antimicrobial resistant microorganisms in both veterinary and human medicine [4,38]. This acquired resistance occurs not only in pathogenic bacteria, but also in the endogenous flora of exposed animals and humans. In poultry, the antimicrobial selection pressure for bacterial drug resistance is quite high with their fecal flora containing a relatively high proportion of resistant bacteria [39]. This could be a possible explanation for high prevalence of MDR *E. coli* observed in our study especially among humans working on poultry farms when compared to those working in the live bird markets where antimicrobial use is minimal. However we were not able to compare our findings with other African countries due to paucity of reports on similar studies.

Factors associated with multidrug resistant *E. coli* in our study were absence of lavatory, poultry worker's history of diarrhea in last three months and occupational exposure of over 10years. This is consistent with findings from a similar study done on Dutch broiler farms that reported occupational exposure to antimicrobials may be responsible to increased resistance rates in humans [24,29,40]. Our study reported resistance to commonly used antimicrobials in poultry among apparently healthy poultry workers. This implies that the observed resistance in our study may have been as a result of exposure of poultry farmers to antimicrobials used on the farm for prophylaxis or treatment of poultry diseases. There are very few studies which are focused on risk factors for multidrug resistant *E. coli* in similar settings. Some studies

reported risk factors such as travel history and use of antimicrobials by the workers themselves however our study did not investigate these factors [40].

## Conclusion

Prevalence of MDR *E. coli* was highest among farm-workers and associated with older farms or markets and poor hygienic measures. Risk factors for acquiring multidrug resistant *E. coli* included absence of lavatory on farms or in the markets, poultry worker's history of diarrhea in last three months and occupational exposure of over 10years. Based on findings from this study, we educated affected farm-workers and poultry-sellers on the importance of hand-hygiene and responsible use of antimicrobials in poultry production. Our study adds to the increasing body of evidence supporting the occupational risks for individuals working on poultry farms or in live bird markets. We recommended to the Municipal and Kuje Area Councils to provide lavatories for public use in farm settlements and markets. The government also needs to enforce prescription of antibiotics for use in poultry production. It is also important that further longitudinal based studies be conducted to explore other possible risk factors for resistant *E. coli* among farm-workers.

## Supporting information

**S1 Appendix. Questionnaire.** Survey of Risk factors for Multi-drug resistant *E. coli* in Poultry and Poultry workers in FCT, North Central Nigeria.
(PDF)

**S2 Appendix. Data set.** Values used for the data analysis.
(XLSX)

**S3 Appendix. Laboratory protocol.**
(PDF)

## Acknowledgments

The authors would like to appreciate Dr. Chikwe Ihekweazu, Director General, Nigeria Center for Disease Control (NCDC), Abuja for providing laboratory space and support towards this research at the NCDC National Reference Laboratory, Gaduwa, Abuja. The authors also acknowledge the intellectual contributions of Dr. Adebayo Adedeji, Miss Eme Ekeng, Mr. Muftan Akinpelu and Mr. Chris Chukwu of the Bacteriology Laboratory at NCDC National Reference Laboratory, Gaduwa, Abuja, towards the success of this research.

## Author Contributions

**Conceptualization:** Mabel Kamweli Aworh, Jacob Kwaga.

**Data curation:** Mabel Kamweli Aworh, Nwando Mba.

**Formal analysis:** Mabel Kamweli Aworh.

**Funding acquisition:** Mabel Kamweli Aworh.

**Investigation:** Mabel Kamweli Aworh, Nwando Mba.

**Methodology:** Mabel Kamweli Aworh.

**Project administration:** Jacob Kwaga, Emmanuel Okolocha.

**Resources:** Mabel Kamweli Aworh.

**Supervision:** Jacob Kwaga, Emmanuel Okolocha, Nwando Mba, Siddhartha Thakur.

**Validation:** Jacob Kwaga.

**Writing – original draft:** Mabel Kamweli Aworh.

**Writing – review & editing:** Jacob Kwaga, Emmanuel Okolocha, Siddhartha Thakur.

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
