## [Decision Letter · Decision Letter 0]

6 Sep 2019

PONE-D-19-19300

Prevalence and Risk Factors for Multi-Drug Resistant Escherichia coli among Poultry Workers in the Federal Capital Territory, Nigeria

PLOS ONE

Dear DR AWORH,

Thank you for submitting your manuscript to PLOS ONE. After careful consideration, we feel that it has merit but does not fully meet PLOS ONE’s publication criteria as it currently stands. Therefore, we invite you to submit a revised version of the manuscript that addresses the points raised during the review process.

We would appreciate receiving your revised manuscript by Oct 21 2019 11:59PM. To enhance the reproducibility of your results, we recommend that if applicable you deposit your laboratory protocols in protocols.io, where a protocol can be assigned its own identifier (DOI) such that it can be cited independently in the future. For instructions see: http://journals.plos.org/plosone/s/submission-guidelines#loc-laboratory-protocols

We look forward to receiving your revised manuscript.

Kind regards,

Grzegorz Woźniakowski, PhD ScD

Academic Editor

PLOS ONE

Journal Requirements:

1. In your Methods section, please provide additional location information, including geographic coordinates for the data set if available.

2. Please include copies of the survey questions or questionnaires used in the study, in both the original language and English, as Supporting Information, or include a citation if they have been published previously.

Reviewers' comments:

Reviewer's Responses to Questions

**Comments to the Author**

1. Is the manuscript technically sound, and do the data support the conclusions?

Reviewer #1: Partly

Reviewer #2: Yes

2. Has the statistical analysis been performed appropriately and rigorously? 

Reviewer #1: Yes

Reviewer #2: Yes

3. Have the authors made all data underlying the findings in their manuscript fully available?

Reviewer #1: Yes

Reviewer #2: No

4. Is the manuscript presented in an intelligible fashion and written in standard English?

Reviewer #1: Yes

Reviewer #2: Yes

5. Review Comments to the Author

Reviewer #1: I think that this is a local issue not a global.

English needs intensive revision.

Authors should clearly mention the study hypothesis in the introduction.

Reviewer #2: Page 4 Line 90: ESBL should be written in full

Page 4 Lines 91-92. Authors should spell out the socioeconomic and public health implications

Page 5 Line 116: Spell out FCT

Page 5 Line 122: Census data from 2006 is too old. Needs an update

Page 6 Sampling: the total number of individuals sampled should be stated here

Page 7 Line 168 Delete (1.0g)

Consistency is needed in referencing Oxiod UK throughout the text

Authors should write in full the first time before using acronyms. A problem throughout the text: Page 8: Line 190 CLSI

Statistical Analysis:

Authors should remove the female record and reanalyze the data.

Authors should address how antibiotics are used in poultry in Nigeria: As prophylaxis, treatment of diseases or growth promotion?

What antibiotics are included in the feed if any?

Conflict of interest from the authors were not declared

Statement of availability of data was not made

6. PLOS authors have the option to publish the peer review history of their article (what does this mean?). If published, this will include your full peer review and any attached files.

Reviewer #1: No

Reviewer #2: No

---

## [Author Response · Author response to Decision Letter 0]

25 Oct 2019

25 October 2019

The Academic Editor,

PLOS ONE Journal.

Dear Editor, 

This is to acknowledge the effort of the editorial team for their immense contributions towards transforming the manuscript to its present form. All the observations raised and the corrections as well as comments made by the reviewers are appreciated and have been addressed. 

Moreover, point-by-point response to reviewers’/editorial comments have been addressed as described in the appendix. 

Should you require additional information or if you have any questions, please feel free to contact me. I will be pleased to hear from you. 

Yours sincerely,

Dr. Mabel K. Aworh (Corresponding author)

mabelaworh@yahoo.com, maworh@ncsu.edu

+234 -803 237 7831, 919 741 1749

 

Reviewer/Editorial Notes and Responses

S/N Comments Authors’ response

1. Journal Requirements

Please ensure that your manuscript meets PLOS ONE's style requirements, including those for file naming. The PLOS ONE style templates can be found at: Many thanks for your feedback. This has been addressed and the manuscript has been revised to ensure it meets PLOS ONE’s style requirements using the templates provided.

2. In your Methods section, please provide additional location information, including geographic coordinates for the data set if available We appreciate these comments of the editorial staff. This has been addressed in the Methods section with the following statements. 

This section now reads: 

This study was conducted in Abuja Municipal Area Council (9.0612° N, 7.4224° E) and Kuje Area Council (8° 52' 46.27" N, 7° 13' 39.22" E) with the largest poultry population of approximately 3.812,288 [18].

Kindly refer to page 6, lines 157 – 160.

3. Please include copies of the survey questions or questionnaires used in the study, in both the original language and English, as Supporting Information, or include a citation if they have been published previously. This comment by the editorial staff is noted. Based on the editorial staff’s comment, a copy of the questionnaire has been provided as supplemental file.

Kindly refer to Supporting Information section, page 25, lines 631 – 632

4. Reviewer #1: I think that this is a local issue not a global.

 This is well received and has been addressed under Introduction section. 

This section now reads: 

In developing countries including Nigeria, many families depend on poultry production as a means of income and livelihood due to the increased consumption of poultry products [5]. Antimicrobial drugs are also readily available over the counter without prescription because access to veterinary drugs is presently not being regulated in the country, thereby encouraging the use of these drugs by poultry farmers indiscriminately in production. Poultry farmers continue to use antibiotics in poultry feed or water for prophylaxis, treatment of diseases and as growth promoters in Nigeria [6,7]. Commonly used antibiotics in poultry production in Nigeria include; oxytetracycline, neomycin, enrofloxacin, doxycycline, gentamicin, colistin, streptomycin, tylosin, ciprofloxacin, nitrofurans and chloramphenicol. This is similar to the practice in most countries [7].

Kindly refer to page 4, lines 99 – 108.

5. English needs intensive revision.

 This very important suggestion by the reviewer is appreciated. We have revised the manuscript accordingly.

6. Authors should clearly mention the study hypothesis in the introduction. We quite appreciate the reviewers for their comments in this regard. The introduction section has now been modified to reflect the hypothesis for this study.

This section now reads: 

We hypothesized that poultry harboring drug resistant E. coli aids in the transmission of the pathogen to human workers who are exposed in farms and live bird markets.

Kindly refer to page 5, lines 145 – 146.

7. Reviewer #2: Page 4 Line 90: ESBL should be written in full

 This is well received and has been addressed under Introduction section. Kindly refer to page 4, lines 111 - 112.

8. Page 4 Lines 91-92. Authors should spell out the socioeconomic and public health implications This is well received and has been addressed under Introduction section. 

This section now reads: 

Resistance to commonly used antibiotics has major socioeconomic and public health implications. The socioeconomic implications of AMR include increased cost and duration of treatment while the public health implications include decreased ability to treat common infections resulting in increased human suffering and ultimately death [9–11].

Kindly refer to page 4, lines 114 - 117.

9. Page 5 Line 116: Spell out FCT This is well received and has been addressed under Introduction section. Kindly refer to page 6, line 153.

10. Page 5 Line 122: Census data from 2006 is too old. Needs an update This is well received and has been addressed under Methods section.

This section now reads: 

Based on data obtained from the National Bureau of Statistics, the population projection for FCT for 2019 is 4,464,785 people [17].

Kindly refer to page 6, lines 156 - 157.

11. Page 6 Sampling: the total number of individuals sampled should be stated here This is well received and has been addressed under Methods section.

Kindly refer to page 7, line 188.

12. Page 7 Line 168 Delete (1.0g) This correction has been made. Kindly refer to page 8, line 210.

13. Consistency is needed in referencing Oxiod UK throughout the text This correction has been made. 

Kindly refer to page 9, line 238.

14. Authors should write in full the first time before using acronyms. A problem throughout the text: Page 8: Line 190 CLSI This is well received and has been addressed under Methods section, page 9, line 240 -241.

15. Statistical Analysis:

Authors should remove the female record and reanalyze the data. This is well received and has been addressed under Results section 

This section now reads: 

Of the 122 poultry workers we sampled, 99.2% (n=121) were male. The analysis presented in the results section was based on 121 males whose ages ranged from 18 to 67 years. The mean age of study participants was 30.6 + 9.7years. The majority of workers were married: 54.6% (n=66), had secondary education: 57.9% (n=70) and worked on a poultry farm: 62.0% (n=75) (Table 1). 

Prevalence of Multidrug Resistant E. coli among poultry workers

A total of 48 (39.7%) individuals tested positive for MDR E. coli. A majority of these, 62.5% (n=30) resided in Abuja Municipal Area Council (Fig 1). The highest prevalence was among farm-workers (POR = 2.7, 95% Confidence Interval [CI] =1.3 – 5.7; p = 0.01) when compared to poultry-sellers. 

Kindly refer to pages 10 and 11, lines 282 - 294.

16. Authors should address how antibiotics are used in poultry in Nigeria: As prophylaxis, treatment of diseases or growth promotion? This is well received and has been addressed under Introduction section

This section now reads: 

Poultry farmers continue to use antibiotics in poultry feed or water for prophylaxis, treatment of diseases and as growth promoters in Nigeria [6,7]. Commonly used antibiotics in poultry production in Nigeria include; oxytetracycline, neomycin, enrofloxacin, doxycycline, gentamicin, colistin, streptomycin, tylosin, ciprofloxacin, nitrofurans and chloramphenicol. This is similar to the practice in most countries [7].

 In 2018, Nigeria’s National Agency for Food and Drugs Administration and Control (NAFDAC) issued a ban on the use of antibiotics as growth promoters in animal feeds. 

 Kindly refer to pages 4, lines 104 - 110.

17. What antibiotics are included in the feed if any? This is well received and has been addressed under Introduction section 

This section now reads: 

Commonly used antibiotics in poultry production in Nigeria include; oxytetracycline, neomycin, enrofloxacin, doxycycline, gentamicin, colistin, streptomycin, tylosin, ciprofloxacin, nitrofurans and chloramphenicol. This is similar to the practice in most countries [7].

Kindly refer to page 4, lines 105 - 108.

18. Conflict of interest from the authors were not declared

Statement of availability of data was not made This is well received and has been addressed. 

These sections now read: 

Competing interests

The authors have declared that there are no competing interests.

Data Availability Statement

All relevant data are within the paper and also available as supporting information (S2 Appendix).

Kindly refer to, page 20, lines 492 – 493 (Competing Interests)

and lines 495 – 497 (Statement of Availability of Data)

---

## [Decision Letter · Decision Letter 1]

5 Nov 2019

Prevalence and Risk Factors for Multi-Drug Resistant Escherichia coli among Poultry Workers in the Federal Capital Territory, Abuja, Nigeria

PONE-D-19-19300R1

Dear Dr. AWORH,

We are pleased to inform you that your manuscript has been judged scientifically suitable for publication and will be formally accepted for publication once it complies with all outstanding technical requirements.

With kind regards,

Grzegorz Woźniakowski, PhD ScD

Academic Editor

PLOS ONE

Additional Editor Comments (optional):

Reviewers' comments:

Reviewer's Responses to Questions

**Comments to the Author**

1. If the authors have adequately addressed your comments raised in a previous round of review and you feel that this manuscript is now acceptable for publication, you may indicate that here to bypass the “Comments to the Author” section, enter your conflict of interest statement in the “Confidential to Editor” section, and submit your "Accept" recommendation.

Reviewer #1: All comments have been addressed

Reviewer #2: All comments have been addressed

2. Is the manuscript technically sound, and do the data support the conclusions?

Reviewer #1: Yes

Reviewer #2: Yes

3. Has the statistical analysis been performed appropriately and rigorously? 

Reviewer #1: Yes

Reviewer #2: Yes

4. Have the authors made all data underlying the findings in their manuscript fully available?

Reviewer #1: Yes

Reviewer #2: Yes

5. Is the manuscript presented in an intelligible fashion and written in standard English?

Reviewer #1: Yes

Reviewer #2: Yes

6. Review Comments to the Author

Reviewer #1: No further comments, Authors have addressed all of my comments. I recommend accepting the paper in its current form.

Reviewer #2: The author have revised the manuscript according to the recommendations made. The language is clear and concise.

7. PLOS authors have the option to publish the peer review history of their article (what does this mean?). If published, this will include your full peer review and any attached files.

Reviewer #1: No

Reviewer #2: No

---

## [Editor Report · Acceptance letter]

12 Nov 2019

PONE-D-19-19300R1 

Prevalence and Risk Factors for Multi-Drug Resistant Escherichia coli among Poultry Workers in the Federal Capital Territory, Abuja, Nigeria 

Dear Dr. AWORH:

I am pleased to inform you that your manuscript has been deemed suitable for publication in PLOS ONE. Congratulations! Your manuscript is now with our production department. 

With kind regards,

on behalf of

Prof. Grzegorz Woźniakowski 

Academic Editor

PLOS ONE